# Immune-Related Gene Signatures to Predict the Effectiveness of Chemoimmunotherapy in Triple-Negative Breast Cancer Using Exploratory Subgroup Discovery

**DOI:** 10.3390/cancers14235806

**Published:** 2022-11-25

**Authors:** Olha Kholod, William I. Basket, Jonathan B. Mitchem, Jussuf T. Kaifi, Richard D. Hammer, Christos N. Papageorgiou, Chi-Ren Shyu

**Affiliations:** 1MU Institute for Data Science and Informatics, University of Missouri, Columbia, MO 65212, USA; 2Department of Surgery, School of Medicine, University of Missouri, Columbia, MO 65212, USA; 3Harry S. Truman Memorial Veterans’ Hospital, Columbia, MO 65201, USA; 4Ellis Fischel Cancer Center, University of Missouri, Columbia, MO 65211, USA; 5Department of Pathology & Anatomical Sciences, School of Medicine, University of Missouri, Columbia, MO 65212, USA; 6Department of Medicine, School of Medicine, University of Missouri, Columbia, MO 65212, USA; 7Department of Electrical Engineering & Computer Science, University of Missouri, Columbia, MO 65212, USA

**Keywords:** triple-negative breast cancer, exploratory subgroup discovery, chemoimmunotherapy

## Abstract

**Simple Summary:**

Chemoimmunotherapy combinations have transformed the treatment landscape for patients with triple-negative breast cancer (TNBC). However, the discovery of immune-related biomarkers is needed to optimally identify patients requiring the addition of immune-checkpoint inhibitors (ICIs) to chemotherapy. In this study, we identified immune-related gene signatures via exploratory subgroup discovery algorithm that substantially increase the odds of partial remission for TNBC patients on anti-PD-L1+chemotherapy regimen. We have also uncovered distinct cell populations for TNBC patients with various treatment outcomes. Our framework may result in better risk stratification for TNBC patients that undergo chemoimmunotherapy and lead to overall improvement of their health outcomes in the future.

**Abstract:**

Triple-negative breast cancer (TNBC) is an aggressive subtype of breast cancer with limited therapeutic options. Although immunotherapy has shown potential in TNBC patients, clinical studies have only demonstrated a modest response. Therefore, the exploration of immunotherapy in combination with chemotherapy is warranted. In this project we identified immune-related gene signatures for TNBC patients that may explain differences in patients’ outcomes after anti-PD-L1+chemotherapy treatment. First, we ran the exploratory subgroup discovery algorithm on the TNBC dataset comprised of 422 patients across 24 studies. Secondly, we narrowed down the search to twelve homogenous subgroups based on tumor mutational burden (TMB, low or high), relapse status (disease-free or recurred), tumor cellularity (high, low and moderate), menopausal status (pre- or post) and tumor stage (I, II and III). For each subgroup we identified a union of the top 10% of genotypic patterns. Furthermore, we employed a multinomial regression model to predict significant genotypic patterns that would be linked to partial remission after anti-PD-L1+chemotherapy treatment. Finally, we uncovered distinct immune cell populations (T-cells, B-cells, Myeloid, NK-cells) for TNBC patients with various treatment outcomes. CD4-Tn-LEF1 and CD4-CXCL13 T-cells were linked to partial remission on anti-PD-L1+chemotherapy treatment. Our informatics pipeline may help to select better responders to chemoimmunotherapy, as well as pinpoint the underlying mechanisms of drug resistance in TNBC patients at single-cell resolution.

## 1. Introduction

Triple-negative breast cancer (TNBC) occurs in about 10 to 20% of diagnosed breast cancers and defined by the absence or minimal expression of estrogen receptor (ER), progesterone receptor (PR) and epidermal growth factor receptor 2 (HER2) [1,2]. Due to its aggressive clinical phenotype and limited response to hormonal therapy, one in three TNBC patients will likely to relapse within the first three years of primary diagnosis [3]. Although numerous therapeutic agents have been evaluated for the treatment of early TNBC [4], only Olaparib has been approved for the treatment of the small group of patients with high-risk TNBC harboring germline BRCA1 or BRCA2 pathogenic variants in the adjuvant setting [5]. The emergence of cancer immunotherapy, however, is altering the paradigm in TNBC treatment. 

TNBC, unlike other breast cancer subtypes, has high tumor mutational burden (TMB), which has been correlated with responsiveness to immune checkpoint inhibitors (ICIs) [6]. Indeed, checkpoint inhibition with the anti-PD1 antibody Pembrolizumab has been approved for advanced-stage, PD-L1 positive TNBC due to improved outcomes when combined with frontline chemotherapy [7]. Interestingly, ICIs are more effective in treating TNBC when given early in the course of the disease, which may be a result of immune escape mechanisms emerging as the condition progresses [8]. More recently, results from the KEYNOTE-522 trial indicated that adding checkpoint inhibition in the early stage setting does in fact improve long-term outcomes [9]. However, subgroup analyses did not pinpoint any strongly predictive biomarkers. For example, PD-L1 expression did not distinguish responders from non-responders in the early setting, with both PD-L1-negative and PD-L1-positive patients obtaining a benefit from Pembrolizumab. Moreover, the addition of immunotherapy increased adverse effects (AEs) [10]. In another study—IMPASSION131– the combination of Paclitaxel with the PD-L1 inhibitor Atezolizumab failed to improve progression-free survival (PFS) or overall survival (OS) in TNBC patients [11]. These findings could be due to imbalances in prognostic features or accidental discoveries in a relatively small trial. Therefore, the exploration of immune-related biomarkers is needed to optimally identify patients requiring the addition of ICIs to chemotherapy [12,13].

In this work we determined homogenous TNBC subgroups based on both phenotypic and genotypic parameters using exploratory subgroup mining. We have also identified significant predictors that increase chances of partial remission in TNBC patients on chemoimmunotherapy treatment using multinomial regression model on TNBC scRNA-seq dataset. Lastly, we uncovered distinct immune cell populations (T-cells, B-cells, Myeloid, NK-cells) for TNBC patients with various treatment outcomes. We interpreted our results using biomedical knowledge, including findings from existing clinical trials, immunohistochemistry experiments and functional characterization of specific genes. The proposed informatics pipeline may assist health care professionals in the selection of chemoimmunotherapy responders, as well as determine the underlying causes of drug resistance in TNBC patients at a single-cell level and resolution. 

## 2. Materials and Methods

### 2.1. Data Mapping

In this study we employed two datasets. Each dataset consisted of multiple phenotypic (either categorical or continuous) and genotypic (continuous only) variables. Each categorical variable was labeled based on the National Comprehensive Cancer Network (NCCN) Guidelines in Oncology [14]. For example, relapse-free status was categorized as (1) disease-free or (2) recurred. Continuous variables were converted into categoric variables by grouping values into several categories. For example, normalized gene expression values were categorized as (1) downregulated, (2) upregulated, or (3) non-differentially expressed. 

The first TNBC dataset comprised of 422 patients. These patients were selected from 24 breast cancer studies available at the cBioPortal platform [15]. The final dataset included breast cancer patients based on the following immunohistochemical profile: ER-negative, PR-negative and HER2-negative. This dataset consisted of 12 phenotypic variables, including clinical-pathologic data (age at diagnosis, menopausal status, tumor type, tumor stage, tumor cellularity, histologic grade, TMB), treatment regimen (chemotherapy, radiotherapy, hormone therapy) and survival status (overall survival status, relapse-free status). There were 1067 genotypic variables in the form of normalized gene expression values derived from human immunome (immune-related genes) and human kinome (protein kinase genes).

The second TNBC dataset consisted of scRNA-seq profiles for 22 TNBC patients that underwent chemotherapy (*Pactilaxel*) or chemoimmunotherapy treatment (*Paclitaxel* with *Atezolizumab*) [16]. For this study we selected six phenotypic variables, including information about treatment timeline (pre-, post-treatment, progression), tissue type (tumor or blood), tumor site (brain, breast, chest wall, liver, lymph nodes), treatment type (anti-PD-L1+chemotherapy or chemotherapy only), treatment response (partial response (PR), stable disease (SD), progressive disease (PD) and cell cluster (T-cells, B-cells, Myeloid, NK-cells)). We used the same genotypic variables as in the TNBC subgroup discovery dataset. 

### 2.2. The Informatics Pipeline

Our informatics pipeline has three modules: (1) exploratory subgroup discovery, (2) inference module based on multinomial regression model and (3) immune cell populations discovery. Our goal was two-fold: (1) to identify significant genes from exploratory subgroup discovery that increase odds of having partial remission after anti-PD-L1+chemotherapy treatment and (2) to uncover unique immune cell populations for TNBC subgroups with various treatment outcomes.

The main goal of exploratory subgroup discovery module was to determine homogenous patient subgroups based on expanatory phenotypic characteristics (Module A on Figure 1), where prevailing number of patients in that subgroup exemplify distinctive genotypic patterns (Module B on Figure 1). Each genotypic pattern had been represented as a combination of differentially expressed genes. For example, the genotypic pattern may consist of three genes: upregulated EGFR, downregulated MTOR and upregulated MAPK1 genes. On the first step, the algorithm determines the base subgroup (e.g., Chemotherapy = Yes) contingent on the most significant contrast against the rest of the population. On the next inclusion step it adds a new phenotypic variable, e.g., TMB = High, to the previous subgroup to generate a more focused subgroup (Chemotherapy = Yes and TMB = High). Subsequently, the exclusion step is employed to remove a less relevant inclusion move after each inclusion step. The exploratory search selects multiple paths that form multiple subgroups and have equally relevant genotypic patterns within each subgroup. When the algorithm reaches the most focused subgroup with the highest contrast score that cannot be further increased, the search would be terminated. Support [17] and growth rate [18] were used to measure the frequency for a specific genotypic pattern in the homogenous subgroup. We then applied a J-value [19] to prioritize each subgroup based on the relevance (contrasts) for all patterns in each subgroup [20]. 

To find significant predictors of partial remission on anti-PD-L1+chemotherapy regimen, we employed multinomial regression model (Module C on Figure 1) on the scRNA-seq TNBC dataset. The outcome variable was categorical and represented as a combination of treatment response, treatment timeline, and treatment type. For example, the level of outcome variable can be encoded as *SD-Post_treatment-Chemo* meaning that a fraction of TNBC patients achieved stable disease after treatment with chemotherapy only. Overall, there were ten levels of outcome variable. We set *PD-Post_treatment-Chemo*—progressive disease after chemotherapy—as a baseline for the model. The continuous covariates were encoded as genes with normalized gene expression values identified as a top 10% of genotypic patterns in the exploratory subgroup discovery stage. We used the multinom function from the nnet package [21] to estimate a multinomial logistic regression model. We computed *p*-values via two-tailed z-test to identify significant predictors of response to anti-PD-L1+chemotherapy treatment. 

The immune cell populations discovery module (Module D on Figure 1) determined distinct immune cell populations (T-cells, B-cells, Myeloid, NK-cells) for TNBC patients with various treatment outcomes. Each TNBC subgroup had two conditions: (1) anti-PD-L1+chemotherapy, post treatment, partial remission and (2) chemotherapy, post treatment, PD. Using the top 10% of genotypic patterns from exploratory mining stage as an input, we generated heatmap plots for each condition in every TNBC subgroup of interest. For example, NME3 gene was represented as a geometric mean of NME3 expression values in CD4-Tcm-LMNA cells [22]. Finally, we compared immune cell populations in these two conditions to identify mutually exclusive cell populations that were associated with either partial remission after anti-PD-L1+chemotherapy treatment or progressive disease after chemotherapy treatment. 

## 3. Results

### 3.1. The Identification of Homogenous TNBC Subgroups

First, we ran the exploratory subgroup discovery algorithm on the TNBC dataset described in Section 2.1. The algorithm revealed 11,944 subgroups. We focused our analysis of the 460 subgroups where TNBC patients had undergone chemotherapy. On the next step, we narrowed down the search to twelve homogenous subgroups based on TMB (low or high), relapse status (disease-free or recurred), tumor cellularity (high, low and moderate), menopausal status (pre- or post) and tumor stage (I, II and III). Since the lengths of genotypic patterns vary (up to 5 genes), we decided to make a union of top 10% of genotypic patterns for each subgroup of interest. Let us assume that each genotypic pattern is a set of elements, where each element is a unique differentially expressed gene (e.g., upregulated *MTOR* gene). The union would represent a set of a collection of genotypic patterns, where each element would not be repetitive. These genotypic patterns were used as covariates for the multinomial regression model in the next section. 

### 3.2. Significant Predictors of Partial Remission after Anti-PD-L1+Chemotherapy 

The multinomial regression model on scRNA-seq TNBC dataset was able to identify significant predictors from exploratory subgroup discovery results that increase odds of having partial remission after anti-PD-L1+chemotherapy treatment versus progressive disease after chemotherapy (Table 1). 

Next, we highlight the importance of identified phenotypic features from Table 1 for TNBC patient outcomes. Using literature, high-TMB TNBC status may benefit specifically from ICIs in combination with chemotherapy [23] or ICIs alone [24]. TNBC patients have high TMB due to accumulation of genomic instability, which leads to the production neoantigens, thereby resulting in strong effector cell responses [25]. TNBC tumors have a “hot tumor phenotype”, which characterized by a high degree of immune infiltration and associated with improved survival outcomes regardless of tumor stage, molecular subtype, PD-L1 status, age and treatment schedule [26]. The IMpassion130 trial tested immunotherapy agent Durvalumab in combination with chemotherapy or chemotherapy alone on 149 early stage TNBC patients. Median TMB was significantly higher in patients with pathologic complete response (pCR) (median 1.87 versus 1.39, *p* = 0.005), and odds ratios for pCR per mut/MB were 2.06 (95% CI 1.33–3.20) among all patients, 1.77 (95% CI 1.00–3.13) in the Durvalumab arm, and 2.82 (95% CI 1.21–6.54) in the chemotherapy arm. Interestingly, the association between pCR and TMB was more pronounced in patients treated with chemotherapy alone. The KEYNOTE-119 trial evaluated metastatic TNBC patients treated with Pembrolizumab monotherapy versus chemotherapy. The positive association was observed between TMB and clinical response to Pembrolizumab (ORR *p* = 0.154, PSF *p* = 0.014, OS *p* = 0.018) but not to chemotherapy (ORR *p* = 0.114, PFS *p* = 0.478, OS *p* = 0.906). ORR and hazard ratio (HR) for OS also suggested a trend towards increased benefit with Pembrolizumab versus chemotherapy in TNBC patients with high TMB. This clinical trial was constrained by the small sample size and low number of TMB-high cases.

In terms of relapse status, one study suggested that rapid versus late relapse in TNBC might be characterized by unique clinical and genomic features [27]. Both ‘rapid relapse’ (rrTNBC) and ‘late relapse’ (lrTNBC) groups had significantly lower expression of immune-related genes. Intriguingly, lrTNBCs were enriched for luminal signatures. There was no difference in TMB or percent genome altered across investigated subgroups of TNBC patients.

In connection to menopausal status, TNBC was observed primarily in postmenopausal patients [28]. The overexpression of the p53 protein, a significantly higher Ki-67 proliferation index value, and a higher nuclear grade was detected in TNBC premenopausal patients. A multivariate analysis estimated that menopausal status, nodal status, and tumor size were significant contributors for disease-free survival (DFS) in TNBC cases.

We had also discovered novel phenotypic features in TNBC subgroups, such as tumor cellularity and tumor stage. The evaluation of tumor cellularity, defined as the percentage of invasive tumor comprised of tumor cells, may represent an informative histologic measure of the differential response of TNBC to chemoimmunotherapy. To classify the severity of a malignant disease in a particular patient, the tumor staging system is employed during the course of disease. This system is essential in optimizing cancer patients treatment options and their risk stratification. Therefore, these features can be important in the design and analysis of intervention studies, including randomized clinical trials, to better assess their prognostic utility for TNBC patients.

### 3.3. Differences in Immune Cell Populations for Discovered TNBC Subgroups

This section described immune cell populations that were discovered in scRNA-seq TNBC data based on genotypic patterns from exploratory mining stage. We interpreted our results using biomedical knowledge, including findings from existing clinical trials, immunohistochemistry experiments and functional characterization of specific genes. The summary of our findings is presented in Table 2.

#### 3.3.1. T-Cells Global Cluster

The proliferative MKI67^+^ T-cells (Tprf-MKI67) were exclusively present in TNBC patients achieving progressive disease after chemotherapy. Based on literature findings, the expression of MKI67 gene was significantly correlated with lymph node metastases, tumor invasion and adverse survival outcome in TNBC [29]. In addition, more unfovourable survival outcomes in breast cancer patients with recurrent lesions were significantly correlated with high Ki-67 immunohistochemical expression levels (hazard ratio 2.307; 95% confidence interval 1.207–4.407, *p*-value = 0.011) [30]. Therefore, MKI67 may be an important biomarker of predictive and prognostic value in TNBC.

CD4-Tn-LEF1 and CD4-CXCL13 T-cells were linked to partial remission after anti-PD-L1+chemotherapy treatment. Importantly, these CD4^+^ T-cells express very high amounts of PD-1 and other co-stimulatory and inhibitory receptors. Therefore, they instrumental to B-cells for efficient antibody responses and their presence in tumor samples is often correlated with a better outcome in patients with solid tumors [31]. Based on biomedical literature, the presence of CD4-CXCL13 T-cells in TNBC tumors responsive to chemoimmunotherapy was detected through immunohistochemistry staining [16,32]. In addition to CXCL13^+^ T-cells, naïve LEF1^+^ T-cells (Tn-LEF1) were also linked to a favorable response to both anti-PD-L1+chemotherapy and chemotherapy. In a recent study, the magnitude of lymphocytic infiltration was assessed by a four-gene signature—HLF, CXCL13, SULT1E1 and GBP1, which was indicative of favourable outcome in TNBC after neoadjuvant therapy. This signature may help to identify early stage TNBC patients and being a novel prognostic biomarker of this aggressive disease [33].

The activated IFI6^+^ T-cells (Tact-IFI6) were linked to progressive disease after chemotherapy. The poor metastasis-free survival in breast cancer patients was linked to upregulation of mitochondrial antiapoptotic protein IFI6 that might be involved in regulation of mitochondrial ROS production [34]. Therefore, to improve clinical outcomes in breast cancer patients, the deactivation of mitochondrial functions of IFI6 is paramount.

#### 3.3.2. B-Cells Global Cluster

The MKI67^+^ follicular B-cells (Bfoc-MKI67), NEIL1^+^ follicular B-cells (Bfoc-NEIL1) and MKI67^+^ memory B-cells (Bmem-MKI67) were exclusively present in TNBC patients with partial remission after anti-PD-L1+chemotherapy treatment. Based on biomedical literature, follicular B-cells was associated with favorable outcomes for TCGA patients with breast cancer [35]. The naïve B-cells, memory B-cells and follicular B-cells were present primarily in patients responsive to chemoimmunotherapy but not in patients responsive to chemotherapy treatment [16]. In regard with Bfoc-NEIL1 cell population, NEIL1 implicated in repair of oxidative damage associated with DNA replication or transcription [36]. Reduction in NEIL1 expression was associated with a poorer outcome in patients with breast invasive carcinoma [37]. Hence, NEIL1 could be a promising biomarker for TNBC patients that consider chemoimmunotherapy treatment.

Plasma IGHG1^+^ B-cells (pB-IGHG1) were linked to progressive disease after chemotherapy treatment. In TNBC, the expression of IGHG1 indicated the most significant prognostic value compared to trivial clinicopathological parameters [38]. Intriguingly, IGHG1 expression in B-cells and plasma cells could be associated with immune evasion and tumor cell proliferation in breast malignancies [39]. These data may imply that B cells or plasma cells could have pro-tumoral roles under particular conditions; however, the factors influencing the emergence of this pathologic phenotype and the roles played by B cells and plasma cells in these contexts remains unclear.

#### 3.3.3. Myeloid Cells Global Cluster

The MMP9^+^ macrophages (macro-MMP9) were exclusively present in TNBC patients with partial remission after anti-PD-L1+chemotherapy treatment. The literature search revealed that MMPs have a intricate role in cancer progression and may exert both pro- and antitumorigenic activities [40]. Although MMP expression has been linked to tumor progression in various cancer types including breast cancer [41], clinical trials investigating the effect of broad-spectrum MMP inhibitors have failed, and in some cases, patients treated with these inhibitors even progressed after treatment comparing to control placebo group [42]. Indeed, the overexpression of MMP9 results in increased production of antiangiogenic fragments, decreased angiogenesis, and therapeutic effects of established breast cancer [43]. In another study, gene transfer of MMP-9 to ex vivo breast cancer tumors caused tumor regression via increased neutrophil infiltration and an activation of tumor-associated macrophages (TAMs) [44]. Therefore, MMP9 can serve as a biomarker for predicting tumor regression in TNBC.

The macro-CCL2, macro-CX3CR1, macro-IFI27, macro-IGFBP7, macro-IL1B9, macro-MGP and macro-SLC40A1 cells were exclusively present in TNBC patients achieving progressive disease after chemotherapy. Based on biomedical findings, CCL2 expression in breast carcinomas was highly associated with macrophage infiltration, and its expression was correlated with poor prognosis in breast cancer patients [45]. In another study, chemokine receptor CX3CR1 showed a role in angiogenic macrophage survival in the tumor microenvironment contributing to tumor metastasis [46]. In a similar fashion, IFI27 overexpression was shown to impair the tamoxifen-induced apoptosis in breast cancer cells [47]. Finally, IL1B signalling contributed to breast cancer metastasis by enhancing tumor cell motility and inhibiting cell proliferation [48]. These findings highlight the importance of CCL2, CX3CR1, IFI27 and IL1B expressed in macrophages in progression of TNBC.

The CLEC9A^+^ dendritic cells (cDC1-CLEC9A), macro-CFD, macro-FOLR2, macro-MKI67, macro-SPP1, macro-TUBA1B, FCN1^+^ monocytes (mono-FCN1), mono-S100A89 and mono-SMIM25 cells were linked to progressive disease after chemotherapy treatment. Notably, CFD functioned as an enhancer of tumor proliferation and cancer stem cell properties in breast cancers [49]. In another study, SPP1-associated macrophages in the tumor-adipose microenvironment facilitate breast cancer progression [50]. Interestingly, S100A8/A9, which are calcium-binding proteins that are secreted primarily by granulocytes and monocytes, may be associated with the loss of estrogen receptor and may be involved in the poor prognosis of Her2^+^/basal-like subtypes of breast cancer. Therefore, myeloid cell populations expressing CFD, SPP1 and S100A89 might be crucial biomarkers of poor treatment response in TNBC.

#### 3.3.4. NK-Cells Global Cluster

The CNOT2^+^ group 2 innate lymphoid cells (ILC2-CNOT2) were exclusively present in TNBC patients achieving partial remission after anti-PD-L1+chemotherapy treatment. Indeed, ILC2s involved in both anti-tumor and pro-tumoral immunity in a variety of human cancers [51]. In terms of pro-tumoral immunity, the promotion of tumor growth and metastasis is achieved by crosstalk between ILC2s and tumor microenviroment (TME) [52]. In addition, the ILC2s trigger the apoptosis of tumor cells by recruiting and activating eosinophils [53], CXCL1L/CXCL2L molecules and macrophages with M1 profile [54].

The ZNF683^+^ group 1 innate lymphoid cells (ILC1-ZNF683) cells were exclusively present in TNBC patients achieving progressive disease after chemotherapy. The biomedical literature demonstrates that ILC1 cells involved in inhibiting the antitumoral immune response, enabling the differential tumor infiltration of ILC1 cells in patients to improve the levaraging of immunity in cancer therapies [55]. However, the role of ZNF683 gene in particular remains elusive.

ILC3-AREG and ILC3-IL7R cells were linked to partial remission after anti-PD-L1+chemotherapy treatment. It had been shown that ILC3-IL7R could predict a favorable response to both treatment regimens, indicating its potential role in effective antitumor immunity [16]. In contrary, ILC1-VCAM1 cells were linked to progressive disease after chemotherapy treatment. Recent studies have shown that vascular cell adhesion molecule-1 (VCAM1) is aberrantly expressed in breast cancer cells and mediates prometastatic tumor-stromal interactions [56]. Therefore, AREG^+^, IL7R^+^ and VCAM1^+^ innate lymphoid cells can help determine prognosis for breast cancer patients.

## 4. Discussion

The analysis of the TNBC scRNA-seq data revealed distinct immune cell populations that are linked to either partial remission after anti-PD-L1+chemotherapy or progressive disease after chemotherapy only. In terms of T-cells, CD4-Tn-LEF1 and CD4-CXCL13 T-cells were linked to partial remission after anti-PD-L1+chemotherapy treatment, while Tact-IFI6 T-cells were linked to progressive disease after chemotherapy. The naïve B-cells, memory B-cells and follicular B-cells were mainly enriched in tumors responsive to chemoimmunotherapy but not in tumors responsive to chemotherapy treatment. The MMP9^+^ macrophages (macro-MMP9) were exclusively present in TNBC patients with partial remission after anti-PD-L1+chemotherapy treatment, while heterogenous population of macrophages, including macro-CCL2, macro-CX3CR1, macro-IFI27, macro-IGFBP7, macro-IL1B9, macro-MGP and macro-SLC40A1 cells were exclusively present in TNBC patients achieving progressive disease after chemotherapy. Finally, group 3 innate lymphoid cells (ILC3-AREG and ILC3-IL7R) were linked to partial remission after anti-PD-L1+chemotherapy treatment, while ZNF683^+^ group 1 innate lymphoid cells (ILC1-ZNF683) cells were exclusively present in TNBC patients achieving progressive disease after chemotherapy. Each of these cell populations have distinctive genetic markers that could be useful therapeutic targets for chemoimmunotherapy.

The role of T follicular helper and B-cell crosstalk in tumor immunity has been extensively studied over the last decade. Accumulating evidence suggests that tumor infiltrated lymphocyte (TIL) subpopulations (CD4, CD8, and CD19/20) constitute of both suppressive (pro-tumor) or effector (anti-tumor) phenotypes whose functions are influenced by the surrounding TME [57]. Natural or treatment-induced immune activation or suppression may determine the balance between pro- or anti-tumor immune cell crosstalk within a given tumor. Key anti-tumor effector activities include antibody-dependent cell cytotoxicity, complement activation, antibody-mediated tumor cell phagocytosis, antigen presentation, T cell activation, cytokine secretion, and direct tumor killing by TIL, including CD8, NK, B cells, and/or macrophages [58].

Despite of significant survival advantages that could be achieved after treatment with chemoimmunotherapy, most TNBC patients would not benefit. Therefore, more and more attention has been paid to the identification and development of biomarkers for the response of chemoimmunotherapy in recent years. Our informatics pipeline identified novel phenotypic and genotypic predictors in unsupervised manner that indicative of favorable outcome after chemoimmunotherapy. These predictors could be important biomarkers in the design and analysis of intervention studies and ultimately could help to optimize therapy decisions for TNBC patients. In addition, it may help to select better responders to chemoimmunotherapy, as well as pinpoint the underlying mechanisms of drug resistance in TNBC patients at single-cell resolution.

## 5. Conclusions

To tackle patient heterogeneity, chemoimmunotherapy combinations represent a feasible alternative for TNBC patients. However, matching patient subgroups to effective treatments that increase their chance of survival remains a challenging endeavor. In this work, we augmented our exploratory subgroup discovery algorithm to identify TNBC subpopulations that may benefit from chemoimmunotherapy. Specifically, we identified immune-related gene signatures that increased the likelihood of partial remission after anti-PD-L1+chemotherapy regimen versus progressive disease after chemotherapy in TNBC patients. Our novel informatics pipeline identified immune cell populations that associated with various treatment outcomes in TNBC. We also showed the importance of TMB and menopausal status among the investigated TNBC subgroups. The potential limitations include the usage of two disjoint datasets and the absence of outcome variable for immunotherapy outcomes in TCGA datasets. Further validation of our computational results in wet-lab studies would be a significant step toward improving survival outcomes for TNBC patients.

## Figures and Tables

**Figure 1 cancers-14-05806-f001:**
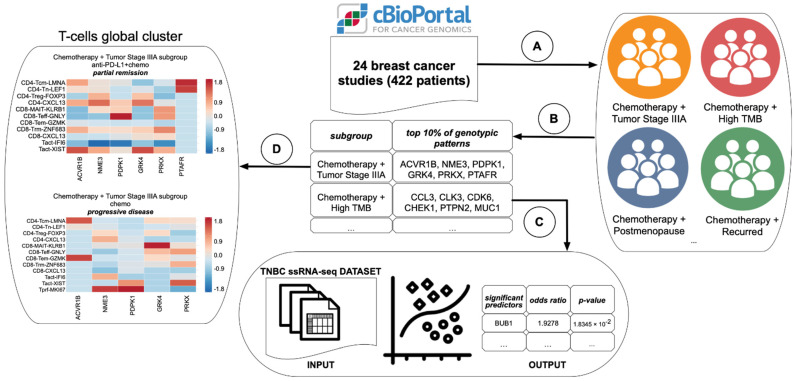
The informatics pipeline. Modules (**A**) and (**B**)—the exploratory subgroup discovery process, Module (**C**)—the inference module based on multinomial regression model, Module (**D**)—immune cell populations discovery.

**Table 1 cancers-14-05806-t001:** Significant predictors that increase odds of partial remission after anti-PD-L1+chemotherapy identified by our informatics pipeline.

Subgroups	Predictors	Coefficients	*p*-Values	Odds Ratio
Chemotherapy (Yes) TMB (High)	ACVR1B	0.4506	0.0136	1.5692
	PDPK1	0.1271	0.0169	1.1355
Chemotherapy (Yes) TMB (Low)	CLK3	0.1402	0.0017	1.1505
	TAOK2	0.6603	<0.0001	1.9354
Chemotherapy (Yes) Relapse Status (Disease Free)	CDK9	0.4152	<0.0001	1.5146
	CFP	0.2368	0.0438	1.2673
	VRK3	0.1591	0.0011	1.1725
Chemotherapy (Yes) Relapse Status (Recurred)	BUB1	0.6563	0.0183	1.9278
	BAZ1B	0.3529	<0.0001	1.4232
Chemotherapy (Yes) Tumor Cellularity (High)	PDIK1L	0.6405	1.3243 × 10^−5^	1.8974
	KIR2DL4	1.8931	0.0094	6.6403
	MAPK3	0.8886	<0.0001	2.4319
	STK24	0.2842	<0.0001	1.3287
Chemotherapy (Yes) Tumor Cellularity (Low)	IFI16	0.0665	0.0205	1.0688
	CSK	0.2397	<0.0001	1.2709
	TAP2	0.2165	0.0185	1.2417
	TIGIT	0.5160	<0.0001	1.6754
Chemotherapy (Yes) Menopausal Status (Pre)	CCR6	0.1307	0.0023	1.1396
	BCL10	0.1222	0.0064	1.1300
	PRKCA	0.9982	<0.0001	2.7135
	EPHB6	0.8559	<0.0001	2.3536
	IFNAR2	0.4546	6.4298 × 10^−6^	1.5756
Chemotherapy (Yes) Menopausal Status (Post)	PDIK1L	0.4952	0.0003	1.6409
	RPS6KA5	0.2836	<0.0001	1.3279
	IKZF2	0.4714	<0.0001	1.6023
Chemotherapy (Yes) Tumor Stage (I)	RIOK3	0.1683	2.4615 × 10^−5^	1.1833
Chemotherapy (Yes) Tumor Stage (II)	PRKCA	1.0213	<0.0001	2.7770
Chemotherapy (Yes) Tumor Stage (III)	IFIH1	0.3366	1.2977 × 10^−5^	0.1773
	CDKL5	0.7786	0.0066	2.1785

**Table 2 cancers-14-05806-t002:** Immune cell populations that are linked to the specific TNBC outcome determined by our informatics pipeline.

Condition	T-Cells	B-Cells	Myeloid Cells	NK-Cells
anti-PD-L1 post treatment partial remission	CD4-Tn-LEF1 CD4-CXCL13	-	-	ILC3-AREG ILC3-IL7R
chemo post treatment progressive disease	Tact-IFI6	pB-IGHG1	cDC1-CLEC9A macro-CFD macro-FOLR2 macro-MKI67 macro-SPP1 macro-TUBA1B mono-FCN1 mono-S100A89 mono-SMIM25	ILC1-VCAM1

## Data Availability

The TCGA dataset is available online at https://www.cbioportal.org (accessed on 22 August 2022). The scRNA-seq TNBC dataset is available online at http://tnbc_pd-l1.cancer-pku.cn (accessed on 22 August 2022).

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
