# Peer review of "Immune-Related Gene Signatures to Predict the Effectiveness of Chemoimmunotherapy in Triple-Negative Breast Cancer Using Exploratory Subgroup Discovery"

_cancers, 2022, doi:10.3390/cancers14235806_

Round 1

Reviewer 1 Report

The authors present a novel algorithm-based informatics pipeline to estimate the impact of immune-related and protein-kinase genotypic patterns on the outcome of chemo-immunotherapy in TNBC. Latter therapy is a hot topic in the field, and the contribution of the study is relevant. As second part, associated immune cell patterns were analyzed, although on a disjoint dataset. A further minor weakness is the absence of an external validation of the results. However, the pipeline performed well on the exploratory data set. Main strengths are appropriate design and novelty. 

The manuscript is clear and firm. Citations are relevant and up-to-date. Data presentation is appropriate, although blending the interpretations and literature contexts with the own results in section 3 (Results) seems to be an extraordinary approach.

Conclusions are consistent with the presented data. The work fit well the scope of Cancers. The conclusions seem to be interesting to the readership of the journal.

Specific comments:

Line 195 (Table 1): Are you sure, if there were any practical benefits using extreme significance levels? This results in extreme low p values (ranging up to 10^-14). Would you consider using a uniform p-value format, for example four or six digits after zero? (Latter method would also refine the probability values given to be literally zero.)

Author Response

Specific comments:

Line 195 (Table 1): Are you sure, if there were any practical benefits using extreme significance levels? This results in extreme low p values (ranging up to 10^-14). Would you consider using a uniform p-value format, for example four or six digits after zero? (Latter method would also refine the probability values given to be literally zero.)

Thank you for your suggestion. We changed small p-values in Table 1 to < 0.0001.

Reviewer 2 Report

Title: Immune-related gene signatures to predict the effectiveness of chemoimmunotherapy in triple-negative breast cancer using exploratory subgroup discovery 

Authors: Olha Kholod , William I. Basket , Jonathan B. Mitchem , Jussuf T. Kaifi , Richard D. Hammer , Christos N. Papageorgiou , Chi-Ren Shyu  

COMMENTS: 

The Authors have dedicated their studies to one of the important problems in the field of cancer treatment, namely the efficacy of combination of chemotherapy and immunotherapy toward triple negative breast cancer (TNBC). After analyses of large datasets including genetic patterns, immune cell populations and state/status of tumor/patients, the Authors have revealed certain groups of genes and certain subpopulations of immune cells that may be used as predictors of outcome of chemotherapy with paclitaxel combined with anti-PD-L1 therapy for patients with TNBC. The obtained results may be helpful for a choice of the correct therapeutic strategy aimed at fighting TNBC and also delineate novel potential targets for attacking this disease.

There are only a couple points of minor criticism: 

1. One of the described Authors' findings is the importance of tumor mutational burden (TMB) for the prediction of outcome of chemoimmunotherapy of TNBC. However, it is generally accepted that checkpoint inhibitor-based immunotherapy is particularly effective against tumors with high TMB. The Authors should better explain (or discuss) how the genetic pattern diversity in TNBC with high and low TMB can define the difference in outcomes of chemoimmunotherapy. 

2. The Authors have also established the significance of patients' menopausal status (pre- or post) for the prediction of outcome of chemoimmunotherapy of TNBC. It seems not quite clear, given that TNBC is the estrogen-independent malignancy. It would be nice if the Authors suggest molecular mechanisms explaining the different responsiveness of TNBC in patients with the different menopausal status to chemoimmunotherapy. 

Author Response

  1. One of the described Authors' findings is the importance of tumor mutational burden (TMB) for the prediction of outcome of chemoimmunotherapy of TNBC. However, it is generally accepted that checkpoint inhibitor-based immunotherapy is particularly effective against tumors with high TMB. The Authors should better explain (or discuss) how the genetic pattern diversity in TNBC with high and low TMB can define the difference in outcomes of chemoimmunotherapy. 

Thank you for your question. TNBC patients have high TMB due to accumulation of genomic instability during disease progression or treatment-associated selective pressure. TNBC tumors have a “hot tumor phenotype”, which characterized by a high degree of immune infiltration and associated with improved survival outcomes regardless of tumor stage, molecular subtype, PD-L1 status, age and treatment schedule. Please, see lines 198 – 203 of the manuscript for our response.

  1. The Authors have also established the significance of patients' menopausal status (pre- or post) for the prediction of outcome of chemoimmunotherapy of TNBC. It seems not quite clear, given that TNBC is the estrogen-independent malignancy. It would be nice if the Authors suggest molecular mechanisms explaining the different responsiveness of TNBC in patients with the different menopausal status to chemoimmunotherapy. 

Thank you for pointing that out. Unfortunately, there is no molecular mechanism explaining the different responsiveness of TNBC in patients with the different menopausal status. The only association described in biomedical literature is between TMB and MMR pathway or homologous recombination repair system deregulation. Please see the following paper: https://pubmed.ncbi.nlm.nih.gov/34229592/.